# Assessing the Scalability of Biologically-Motivated Deep Learning Algorithms and Architectures

**Sergey Bartunov**
DeepMind

**Adam Santoro**
DeepMind

**Blake A. Richards**
University of Toronto

**Luke Marris**
DeepMind

**Geoffrey E. Hinton**
Google Brain

**Timothy P. Lillicrap**
DeepMind, University College London

## Abstract

The backpropagation of error algorithm (BP) is impossible to implement in a real brain. The recent success of deep networks in machine learning and AI, however, has inspired proposals for understanding how the brain might learn across multiple layers, and hence how it might approximate BP. As of yet, none of these proposals have been rigorously evaluated on tasks where BP-guided deep learning has proved critical, or in architectures more structured than simple fully-connected networks. Here we present results on scaling up biologically motivated models of deep learning on datasets which need deep networks with appropriate architectures to achieve good performance. We present results on the MNIST, CIFAR-10, and ImageNet datasets, explore variants of target-propagation (TP) and feedback alignment (FA) algorithms, and examine performance in both fully- and locally-connected architectures. We also introduce weight-transport-free variants of difference target propagation (DTP) modified to remove backpropagation from the penultimate layer. Many of these algorithms perform well for MNIST, but for CIFAR and ImageNet we find that TP and FA variants perform significantly worse than BP, especially for networks composed of locally connected units, opening questions about whether new architectures and algorithms are required to scale these approaches. Our results and implementation details help establish baselines for biologically motivated deep learning schemes going forward.

## 1 Introduction

The suitability of the backpropagation of error (BP) algorithm [32] for explaining learning in the brain was questioned soon after it was popularized [11, 8]. Weaker objections included undesirable characteristics of artificial networks in general, such as their violation of Dale's Law, their lack of cell-type variability, and the need for the gradient signals to be both positive and negative. More serious objections were: (1) The need for the feedback connections carrying the gradient to have the same weights as the corresponding feedforward connections and (2) The need for a distinct form of information propagation (error feedback) that does not influence neural activity, and hence does not conform to known biological feedback mechanisms underlying neural communication. Researchers have long sought biologically plausible and empirically powerful learning algorithms that avoid these flaws [2, 30, 31, 1, 26, 39, 14, 16, 12, 5, 23]. Recent work has demonstrated that the first objection may not be as problematic as often supposed [22]: the feedback alignment (FA) algorithm uses random weights in backward pathways to successfully deliver error information to earlier layers. At the same time, FA still suffers from the second objection: it requires the delivery of signed error vectors via a distinct pathway.

Another family of promising approaches to biologically motivated deep learning – such as Contrastive Hebbian Learning [24], and Generalized Recirculation [26] – use top-down feedback connections to influence *neural activity*, and *differences* in feedfoward-driven and feedback-driven activities (or products of activities) to locally approximate gradients [1, 31, 26, 39, 4, 36, 38]. Since these activity propagation methods don't require explicit propagation of gradients through the network, they go a long way towards answering the second serious objection noted above. However, many of these methods require long "positive" and "negative" settling phases for computing the activities whose differences provide the learning signal. Proposals for shortening the phases [13, 6] are not entirely satisfactory as they still fundamentally depend on a settling process, and, in general, any settling process will likely be too slow for a brain that needs to quickly compute hidden activities in order to act in real time.

Perhaps the most practical among this family of "activity propagation" algorithms is target propagation (TP) and its variants [19, 20, 13, 3, 21]. TP avoids the weight transport problem by training a distinct set of feedback connections that define the backward activity propagation. These connnections are trained to approximately invert the computation of the feedforward connections in order to be able to compute *target activities* for each layer by successively inverting the desired output target. Another appealing property of TP is that the errors guiding weight updates are computed locally along with backward activities.

While TP and its variants are promising as biologically-motivated algorithms, there are lingering questions about their applicability to the brain. First, the only variant explored empirically (i.e. DTP) still depends on explicit gradient computation via backpropagation for learning the penultimate layer's outgoing synaptic weights (see Algorithm Box 1 in Lee et al. [21]). Second, they have not been rigorously tested on datasets more difficult than MNIST. And third, they have not been incorporated into architectures more complicated than simple multi-layer perceptrons (MLPs).

On this second point, it might be argued that an algorithm's inability to scale to difficult machine learning datasets is a red herring when assessing whether it could help us understand learning in the brain. Performance on isolated machine learning tasks using a model that lacks other adaptive neural phenomena – e.g., varieties of plasticity, evolutionary priors, etc. – makes a statement about the lack of these phenomena as much as it does about the suitability of an algorithm. Nonetheless, we argue that there is a need for *behavioural* realism, in addition to *physiological* realism, when gathering evidence to assess the overall *biological* realism of a learning algorithm. Given that human beings are able to learn complex tasks that bear little relationship to their evolution, it would appear that the brain possesses a powerful, general-purpose learning algorithm for shaping behavior. As such, researchers can, and should, seek learning algorithms that are both more plausible physiologically, and scale up to the sorts of complex tasks that humans are capable of learning. Augmenting a model with adaptive capabilities is unlikely to unveil any truths about the brain if the model's performance is crippled by an insufficiently powerful learning algorithm. On the other hand, demonstrating good performance with even a vanilla artificial neural network provides evidence that, at the very least, the learning algorithm *is not limiting*. Ultimately, we need a confluence of evidence for: (1) the sufficiency of a learning algorithm, (2) the impact of biological constraints in a network, and (3) the necessity of other adaptive neural capabilities. This paper focuses on addressing the first two.

In this work our contribution is threefold: (1) We examine the learning and performance of biologically-motivated algorithms on MNIST, CIFAR, and ImageNet. (2) We introduce variants of DTP which eliminate significant lingering biologically implausible features from the algorithm. (3) We investigate the role of weight-sharing convolutions, which are key to performance on difficult datasets in artificial neural networks, by testing the effectiveness of locally connected architectures trained with BP and variants of FA and TP.

Overall, our results are largely negative. That is, we find that none of the tested algorithms are capable of effectively scaling up to training large networks on ImageNet. There are three possible interpretations from these results: (1) Existing algorithms need to be modified, added to, and/or optimized to account for learning in the real brain, (2) research should continue into new physiologically realistic learning algorithms that can scale-up, or (3) we need to appeal to other adaptive capacities to account for the fact that humans are able to perform well on this task. Ultimately, our negative results are important because they demonstrate the need for continued work to understand the power of learning in the human brain. More broadly, we suggest that behavioural realism, as judged by performance

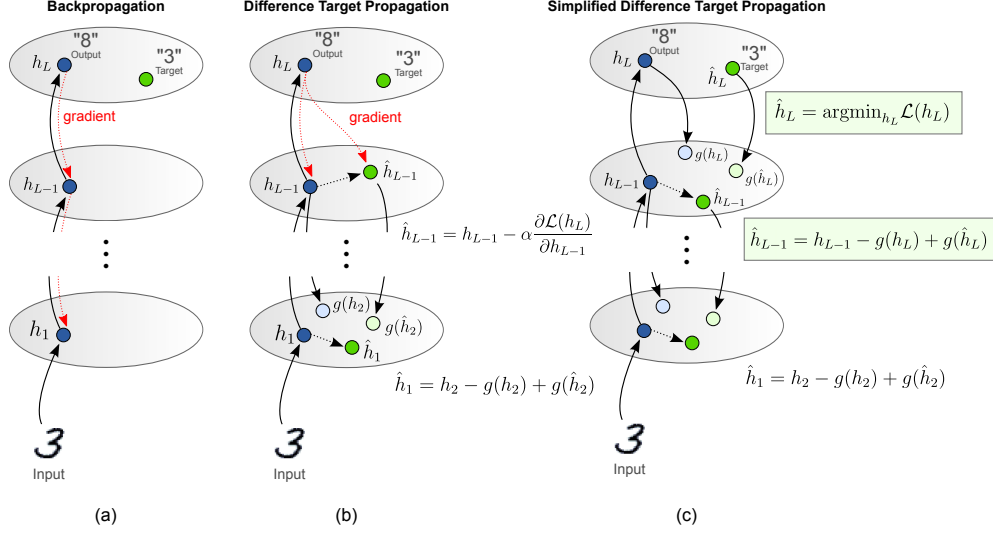

Figure 1: In BP and DTP, the final layer target is used to compute a loss, and the gradients from this loss are shuttled backwards (through all layers, in BP, or just one layer, in DTP) in error propagation steps that do not influence actual neural activity. SDTP never transports gradients using error propagation steps, unlike DTP and BP.

on difficult tasks, should increasingly become one of the metrics used in evaluating the biological realism of computational models and algorithms.

## 2 Learning in Multilayer Networks

Consider the case of a feed-forward neural network with $L$ layers $\{h_l\}_{l=1}^{L}$, whose activations $h_l$ are computed by elementwise-applying a non-linear function $\sigma_l$ to an affine transformation of previous layer activations $h_{l-1}$:

$$h_l = f(h_{l-1}; \theta_l) = \sigma_l(W_l h_{l-1} + b_l), \quad \theta_l = \{W_l, b_l\}, \tag{1}$$

with input to the network denoted as $h_0 = x$ and the last layer $h_L$ used as output.

In classification problems the output layer $h_L$ parametrizes a predicted distribution over possible labels $p(y|h_L)$, usually using the softmax function. The learning signal is then provided as a loss $\mathcal{L}(h_L)$ incurred by making a prediction for an input $x$, which in the classification case can be cross-entropy between the ground-truth label distribution $q(y|x)$ and the predicted one: $\mathcal{L}(h_L) = -\sum_y q(y|x) \log p(y|h_L)$. The goal of training is then to adjust the parameters $\Theta = \{\theta_l\}_{l=1}^{L}$ in order to minimize a given loss over the training set of inputs.

### 2.1 Backpropagation

Backpropagation [32] was popularized as a method for training neural networks by computing gradients with respect to layer parameters using the chain rule:

$$\frac{d\mathcal{L}}{dh_l} = \left(\frac{dh_{l+1}}{dh_l}\right)^T \frac{d\mathcal{L}}{dh_{l+1}}, \quad \frac{d\mathcal{L}}{d\theta_l} = \left(\frac{dh_l}{d\theta_l}\right)^T \frac{d\mathcal{L}}{dh_l}, \quad \frac{dh_{l+1}}{dh_l} = W_{l+1}\text{diag}(\sigma'_{l+1}(W_{l+1}h_l + b_{l+1})).$$

Thus, gradients are obtained by first propagating activations forward to the output layer via eq. 1, and then recursively applying these backward equations. These equations imply that gradients are propagated backwards through the network using weights symmetric to their feedforward counterparts. This is biologically problematic because it implies a mode of information propagation (error propagation) that does not influence neural activity, and that depends on an implausible network architecture (symmetric weight connectivity for feedforward and feedback directions, which is called the weight transport problem).

### 2.1.1 Feedback alignment

While we focus on TP variants in this manuscript, with the purpose of a more complete experimental study of biologically motivated algorithms, we explore FA as another baseline. FA replaces the transpose weight matrices in the backward pass for BP with fixed random connections. Thus, FA shares features with both target propagation and conventional backpropagation. On the one hand, it alleviates the weight transport problem by maintaining a separate set of connections that, under certain conditions, lead to synchronized learning of the network. On the other hand, similar to backpropagation, FA transports signed error information in the backward pass, which may be problematic to implement as a plausible neural computation. We consider both the classical variant of FA [23] with random feedback weights at each hidden layer, and the recently proposed Direct Feedback Alignment [25] (DFA) or Broadcast Feedback Alignment [35], which connect feedback from the output layer directly to all previous layers directly.

### 2.1.2 Target propagation and its variants

Unlike backpropagation, where backwards communication passes on gradients without inducing or altering neural activity, the backward pass in target propagation [19, 20, 3, 21] takes place in the same space as the forward-pass neural activity. The backward induced activities are those that layers should strive to match so as to produce the target output. After feedforward propagation given some input, the final output layer $h_L$ is trained directly to minimize the loss $\mathcal{L}$, while all other layers are trained so as to match their associated targets.

In general, good targets are those that minimize the loss computed in the output layer if they had been realized in feedforward propagation. In networks with invertible layers one could generate such targets by first finding a loss-optimal output activation $\hat{h}_L$ (e.g. the correct label distribution) and then propagating it back using inverse transformations $\hat{h}_l = f^{-1}(\hat{h}_{l+1}; \theta_{l+1})$. Since it is hard to maintain invertibility in a network, approximate inverse transformations (or decoders) can be learned $g(h_{l+1}; \lambda_{l+1}) \approx f^{-1}(h_{l+1}; \theta_{l+1})$. Note that this learning obviates the need for symmetric weight connectivity.

The generic form of target propagation algorithms we consider in this paper can be summarized as a scheduled minimization of two kinds of losses for each layer.

1. *Reconstruction* or *inverse loss* $\mathcal{L}_l^{inv}(\lambda_l) = \|h_{l-1} - g(f(h_{l-1}; \theta_{l-1}); \lambda_l)\|_2^2$ is used to train the approximate inverse that is parametrized similarly to the forward computation: $g(h_l; \lambda_l) = \sigma_l(V_l h_l + c_l), \lambda_l = \{V_l, c_l\}$, where activations $h_{l-1}$ are assumed to be propagated from the input. One can imagine other learning rules for the inverse, for example, the original DTP algorithm trained inverses on noise-corrupted versions of activations with the purpose of improved generalization. The loss is applied for every layer except the first, since the first layer does not need to propagate target inverses backwards.

2. *Forward loss* $\mathcal{L}_l(\theta_l) = \|f(h_l; \theta_l) - \hat{h}_{l+1}\|_2^2$ penalizes the layer parameters for producing activations different from their targets. Parameters of the last layer are trained to minimize the task's loss $\mathcal{L}$ directly.

Under this framework both losses are local and involve only a single layer's parameters, and implicit dependencies on other layer's parameters are ignored. Variants differ in the way targets $\hat{h}_l$ are computed.

**Target propagation**   "Vanilla" target propagation (TP) computes targets by propagating the higher layers' targets backwards through layer-wise inverses; i.e. $\hat{h}_l = g(\hat{h}_{l+1}; \lambda_{l+1})$. For traditional categorization tasks the same 1-hot vector in the output will always map back to precisely the same hidden unit activities in a given layer. Thus, this kind of naive TP may have difficulties when different instances of the same class have different appearances, since it will attempt to make their representations identical even in the early layers. As well, there are no guarantees about how TP will behave when the inverses are imperfect.

**Difference target propagation**   Both TP and DTP update the output weights and biases using the standard delta rule, but this is biologically unproblematic because it does not require weight

transport [26, 23]. For most other layers in the network, DTP [21] computes targets as

$$\hat{h}_l = g(\hat{h}_{l+1}; \lambda_{l+1}) + [h_l - g(h_{l+1}; \lambda_{l+1})]. \tag{2}$$

The second term is the error in the reconstruction, which provides a stabilizing linear correction for imprecise inverse functions. However, in the original work by Lee et al. [21] the penultimate layer target, $\hat{h}_{L-1}$, was computed using gradients from the network's loss, rather than by target propagation. That is, $\hat{h}_{L-1} = h_{L-1} - \alpha \frac{\partial \mathcal{L}(h_L)}{\partial h_{L-1}}$, rather than $\hat{h}_{L-1} = h_{L-1} - g(h_L; \lambda_L) + g(\hat{h}_L; \lambda_L)$. Though not stated explicitly, this approach was presumably taken to ensure that the penultimate layer received reasonable and diverse targets despite the low-dimensional 1-hot targets at the output layer. When there are a small number of 1-hot targets (e.g. 10 classes), learning a good inverse mapping from these vectors back to the hidden activity of the penultimate hidden layer (e.g. 1000 units) might be problematic, since the inverse mapping cannot provide information that is both useful and unique to a particular input sample $x$. Using BP in the penultimate layer sidesteps this concern, but deviates from the intent of using these algorithms to avoid gradient computation and delivery.

**Simplified difference target propagation**    We introduce SDTP as a simple modification to DTP. In SDTP we compute the target for the penultimate layer as $\hat{h}_{L-1} = h_{L-1} - g(h_L; \lambda_L) + g(\hat{h}_L; \lambda_L)$, where $\hat{h}_L = \text{argmin}_{h_L} \mathcal{L}(h_L)$, i.e. the correct label distribution. This completely removes biologically infeasible gradient communication (and hence weight-transport) from the algorithm. However, it is not clear whether targets for the penultimate layer will be diverse enough (given low entropy classification targets) or precise enough (given the inevitable poor performance of the learned inverse for this layer). The latter is particularly important if the dimensionality of the penultimate layer is much larger than the output layer, which is the case for classification problems with a small number of classes. Hence, this modification is a non-trivial change that requires empirical investigation. In Section 3 we evaluate SDTP in the presence of low-entropy targets (classification problems) and also consider the problem of learning an autoencoder (for which targets are naturally high-dimensional and diverse) in the supplementary material.

---

**Algorithm 1** Simplified Difference Target Propagation

---

    Propagate activity forward:
    **for** $l = 1$ **to** $L$ **do**
        $h_l \leftarrow f_l(h_{l-1}; \theta_l)$
    **end for**
    Compute first target: $\hat{h}_L \leftarrow \text{argmin}_{h_L} \mathcal{L}(h_L)$
    Compute targets for lower layers:
    **for** $l = L - 1$ **to** $1$ **do**
        $\hat{h}_l \leftarrow h_l - g(h_{l+1}; \lambda_{l+1}) + g(\hat{h}_{l+1}; \lambda_{l+1})$
    **end for**
    Train inverse function parameters:
    **for** $l = L$ **to** $2$ **do**
        Generate corrupted activity $\tilde{h}_{l-1} = h_{l-1} + \epsilon, \epsilon \sim \mathcal{N}(0, \sigma^2)$
        Update parameters $\lambda_l$ using SGD on loss $\mathcal{L}_l^{inv}(\lambda_l)$
        $\mathcal{L}_l^{inv}(\lambda_l) = \|h_{l-1} - g(f(\tilde{h}_{l-1}; \theta_{l-1}); \lambda_l)\|_2^2$
    **end for**
    Train feedforward function parameters:
    **for** $l = 1$ **to** $L$ **do**
        Update parameters $\theta_l$ using SGD on loss $\mathcal{L}_l(\theta_l)$
        $\mathcal{L}_l(\theta_l) = \|f(h_l; \theta_l) - \hat{h}_{l+1}\|_2^2$ if $l < L$, else $\mathcal{L}_L(\theta_L) = \mathcal{L}$ (task loss)
    **end for**

---

**Auxiliary output SDTP**    As outlined above, in the context of 1-hot classification, SDTP produces only weak targets for the penultimate layer, i.e. one for each possible class label. To circumvent this problem, we extend SDTP by introducing a composite structure for the output layer $h_L = [o, z]$, where $o$ is the predicted class distribution on which the loss is computed and $z$ is an auxiliary output vector that is meant to provide additional information about activations of the penultimate layer $h_{L-1}$. Thus, the inverse computation $g(h_L; \lambda_L)$ can be performed *conditional on richer information from the input*, not just on the relatively weak information available in the predicted and actual label.

The auxiliary output $z$ is used to generate targets for penultimate layer as follows:

$$\hat{h}_{L-1} = h_{L-1} - g_L(o, z; \lambda_L) + g_L(\hat{o}, z; \lambda_L), \qquad (3)$$

where $o$ is the predicted class distribution, $\hat{o}$ is the correct class distribution and $z$ produced from $h_{L-1}$ is used in both inverse computations. Here $g_L(\hat{o}, z; \lambda_L)$ can be interpreted as a modification of $h_L$ that preserves certain features of the original $h_L$ that can also be classified as $\hat{o}$. Here parameters $\lambda_L$ can be still learned using the usual inverse loss. But parameters of the forward computation $\theta_{L-1}$ used to produce $z$ are difficult to learn in a way that maximizes their effectiveness for reconstruction without backpropagation. Thus, we studied a variant that does not require backpropagation: we simply do not optimize the forward weights for $z$, so $z$ is just a set of random features of $h_{L-1}$.

**Parallel and alternating training of inverses**  In the original implementation of DTP[1], the authors trained forward and inverse model parameters by alternating between their optimizations; in practice they trained one loss for one full epoch of the training set before switching to training the other loss. We considered a variant that simply optimizes both losses in parallel, which seems nominally more plausible in the brain since both forward and feedback connections are thought to undergo plasticity changes simultaneously — though it is possible that a kind of alternating learning schedule for forward and backward connections could be tied to wake/sleep cycles.

## 2.2  Biologically-plausible network architectures

Convolution-based architectures have been critical for achieving state of the art in image recognition [18]. These architectures are biologically implausible, however, because of their extensive weight sharing. To implement convolutions in biology, many neurons would need to share the values of their weights precisely — a requirement with no empirical support. In the absence of weight sharing, the "locally connected" receptive field structure of convolutional neural networks is in fact very biologically realistic and may still offer a useful prior. Under this prior, neurons in the brain could sample from small areas of visual space, then pool together to create spatial maps of feature detectors.

On a computer, sharing the weights of locally connected units greatly reduces the number of free parameters and this has several beneficial effects on simulations of large neural nets. It improves generalization and it drastically reduces both the amount of memory needed to store the parameters and the amount of communication required between replicas of the same model running on different subsets of the data on different processors. From a biological perspective we are interested in how TP and FA compare with BP without using weight sharing, so both our BP results and our TP and FA results are considerably worse than convolutional neural nets and take far longer to produce. We assess the degree to which BP-guided learning is enhanced by convolutions, and not BP *per se*, by evaluating learning methods (including BP) on networks with locally connected layers.

# 3  Experiments

In this section we experimentally evaluate variants of target propagation, backpropagation, and feedback alignment [23, 25]. We focused our attention on TP variants. We found all of the variants we explored to be quite sensitive to the choice of hyperparameters and network architecture, especially in the case of locally-connected networks. With the aim of understanding the limits of the considered algorithms, we manually searched for architectures well suited to DTP. Then we fixed these architectures for BP and FA variants and ran independent hyperparameter searches for each learning method. Finally, we report best errors achieved in 500 epochs. For additional details see Tables 3 and 4 in the Appendix.

For optimization we use Adam [15], with different hyper-parameters for forward and inverse models in the case of target propagation. All layers are initialized using the method suggested by Glorot & Bengio [10]. In all networks we used the hyperbolic tangent as a nonlinearity between layers as it was previously found to work better with DTP than ReLUs [21].

Table 1: Train and test errors (%) achieved by different learning methods for fully-connected (FC) and locally-connected (LC) networks on MNIST and CIFAR. We highlight **best** and **second best** results.

| | (a) MNIST | | | | (b) CIFAR | | | |
| | FC | | LC | | FC | | LC | |
| METHOD | TRAIN | TEST | TRAIN | TEST | TRAIN | TEST | TRAIN | TEST |
|---|---|---|---|---|---|---|---|---|
| DTP, PARALLEL | 0.44 | 2.86 | **0.00** | **1.52** | 59.45 | 59.14 | 28.69 | 39.47 |
| DTP, ALTERNATING | **0.00** | **1.83** | **0.00** | **1.46** | **30.41** | **42.32** | 28.54 | 39.47 |
| SDTP, PARALLEL | 1.14 | 3.52 | 0.00 | 1.98 | 51.48 | 55.32 | 43.00 | 46.63 |
| SDTP, ALTERNATING | 0.00 | 2.28 | 0.00 | 1.90 | 48.65 | 54.27 | 40.40 | 45.66 |
| AO-SDTP, PARALLEL | 0.96 | 2.93 | 0.00 | 1.92 | 4.28 | 47.11 | 32.67 | 40.05 |
| AO-SDTP, ALTERNATING | **0.00** | **1.86** | 0.00 | 1.91 | 0.00 | 45.40 | 34.11 | 40.21 |
| FA | **0.00** | **1.85** | **0.00** | **1.26** | 25.62 | **41.97** | 17.46 | 37.44 |
| DFA | 0.85 | 2.75 | 0.23 | 2.05 | 33.35 | 47.80 | 32.74 | 44.41 |
| BP | **0.00** | **1.48** | **0.00** | **1.17** | **28.97** | **41.32** | **0.83** | **32.41** |
| BP CONVNET | – | – | **0.00** | **1.01** | – | – | **1.39** | **31.87** |

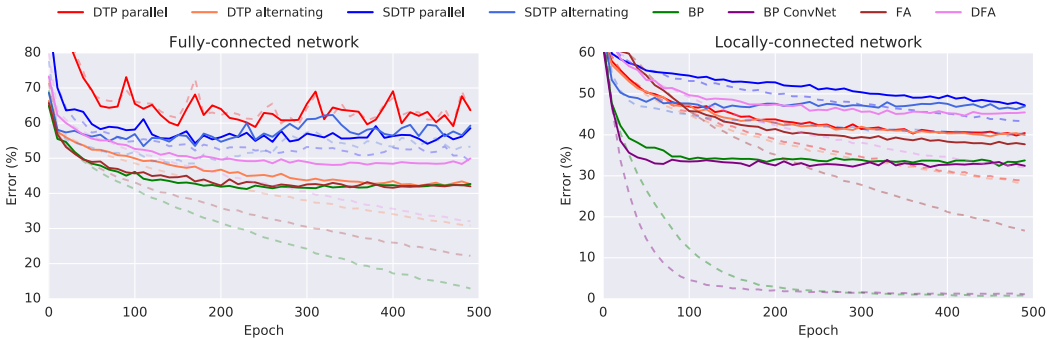

Figure 2: Train (dashed) and test (solid) classification errors on CIFAR.

## 3.1 MNIST

To compare to previously reported results we began with the MNIST dataset, consisting of $28 \times 28$ gray-scale images of hand-drawn digits. The final performance for all algorithms is reported in Table 1 and the learning dynamics are plotted in Figure 8 (see Appendix). Our implementation of DTP matches the performance of the original work [21]. However, all variants of TP performed slightly worse than BP, with a larger gap for SDTP, which does not rely on any gradient propagation. Interestingly, alternating optimization of forward and inverse losses consistently demonstrates more stable learning and better final performance.

## 3.2 CIFAR-10

CIFAR-10 is a more challenging dataset introduced by Krizhevsky [17]. It consists of $32 \times 32$ RGB images of 10 categories of objects in natural scenes. In contrast to MNIST, classes in CIFAR-10 do not have a "canonical appearance" such as a "prototypical bird" or "prototypical truck" as opposed to "prototypical 7" or "prototypical 9". This makes them harder to classify with simple template matching, making depth imperative for achieving good performance. The only prior study of biologically motivated learning methods applied to this data was carried out by Lee et al. [21]; this investigation was limited to DTP with alternating updates and fully connected architectures. Here we present a more comprehensive evaluation that includes locally-connected architectures and experiments with an augmented training set consisting of vertical flips and random crops applied to the original images.

Final results can be found in Table 1. Overall, the results on CIFAR-10 are similar to those obtained on MNIST, though the gap between TP and backpropagation as well as between different variants of TP is more prominent. Moreover, while fully-connected DTP-alternating roughly matched the

performance of BP, locally-connected networks presented an additional challenge for TP, yielding only a minor improvement.

The issue of compatibility with locally-connected layers is yet to be understood. One possible explanation is that the inverse computation might benefit from a form that is not symmetric to the forward computation. We experimented with more expressive inverses, such as having larger receptive fields or a fully-connected structure, but these did not lead to any significant improvements. We leave further investigation of this question to future work.

As with MNIST, a BP trained convolutional network with shared weights performed better than its locally-connected variant. The gap, however, is not large, suggesting that weight sharing is not necessary for good performance as long as the learning algorithm is effective.

We hypothesize that the significant gap in performance between DTP and the gradient-free SDTP on CIFAR-10 is due to the problems with inverting a low-entropy target in the output layer. To validate this hypothesis, we ran AO-SDTP with 512 auxiliary output units and compare its performance with other variants of TP. Even though the observed results do not match the performance of DTP, they still present a large improvement over SDTP. This confirms the importance of target diversity for learning in TP (see Appendix 5.5 for related experiments) and provides reasonable hope that future work in this area could further improve the performance of SDTP.

Feedback alignment algorithm performed quite well on both MNIST and CIFAR, struggling only with the LC architecture on CIFAR. In contrast, DFA appeared to be quite sensitive to the choice of architecture and our architecture search was guided by the performance of TP methods. Thus, the numbers achieved by DFA in our experiments should be regarded only as a rough approximation of the attainable performance for the algorithm. In particular, DFA appears to struggle with the relatively narrow (256 unit) layers used in the fully-connected MNIST case — see Lillicrap et al. [23] Supplementary Information for a possible explanation. Under these conditions, DFA fails to match BP in performance, and also tends to fall behind DTP and AO-SDTP, especially on CIFAR.

### 3.3   ImageNet

We assessed performance of the methods on the ImageNet dataset [33], a large-scale benchmark that has propelled recent progress in deep learning. To the best of our knowledge, this is the first empirical study of biologically-motivated methods and architectures conducted on a dataset of such scale and difficulty. ImageNet has 1000 object classes appearing in a variety of natural scenes and captured in high-resolution images (resized to $224 \times 224$).

Final results are reported in Table 2. Unlike MNIST and CIFAR, on ImageNet all biologically motivated algorithms performed very poorly relative to BP. A number of factors could contribute to this result. One factor may be that deeper networks might require more careful hyperparameter tuning; for example, different learning rates or amounts of noise injected for each layer.

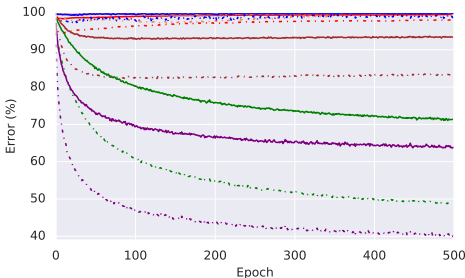

Figure 3: Top-1 (solid) and Top-5 (dotted) test errors on ImageNet. Color legend is the same as for figure 2.

Table 2: Test errors on ImageNet.

| METHOD | TOP-1 | TOP-5 |
|---|---|---|
| DTP, PARALLEL | 98.34 | 94.56 |
| DTP, ALTERNATING | 99.36 | 97.28 |
| SDTP, PARALLEL | 99.28 | 97.15 |
| FA | 93.08 | 82.54 |
| BACKPROPAGATION | **71.43** | **49.07** |
| BACKPROPAGATION, CONVNET | **63.93** | **40.17** |

A second factor might be a general incompatibility between the mainstream design choices for convolutional networks with TP and FA algorithms. Years of research have led to a better understanding of efficient architectures, weight initialization, and optimizers for convolutional networks trained with backpropagation, and perhaps more effort is required to reach comparable results for biologically motivated algorithms and architectures. Addressing both of these factors could help

improve performance, so it would be premature to conclude that TP cannot perform adequately on ImageNet. We can conclude though, that out-of-the-box application of this class of algorithms does not provide a straightforward solution to real data on even moderately large networks.

We note that FA demonstrated an improvement over TP, yet still performed much worse than BP. It was not practically feasible to run its sibling, DFA, on large networks such as one we used in our ImageNet experiments. This was due to practical necessity of maintaining a large fully-connected feedback layer of weights from the output layer to each intermediate layer. Modern convolutional architectures tend to have very large activation dimensions, and the requirement for linear projections back to all of the neurons in the network is practically intractable: on a GPU with 16GB of onboard memory, we encountered out-of-memory errors when trying to initialize and train these networks using a Tensorflow implementation. Thus, the DFA algorithm appears to require either modification or GPUs with more memory to run with large networks.

## 4   Discussion

Historically, there has been significant disagreement about whether BP can tell us anything interesting about learning in the brain [8, 11]. Indeed, from the mid 1990s to 2010, work on applying insights from BP to help understand learning in the brain declined precipitously. Recent progress in machine learning has prompted a revival of this debate; where other approaches have failed, deep networks trained via BP have been key to achieving impressive performance on difficult datasets such as ImageNet. It is once again natural to wonder whether some approximation of BP might underlie learning in the brain [22, 5]. However, none of the algorithms proposed as approximations of BP have been tested on the datasets that were instrumental in convincing the machine learning and neuroscience communities to revisit these questions.

Here we studied TP and FA, and introduced a straightforward variant of the DTP algorithm that completely removed gradient propagation and weight transport. We demonstrated that networks trained with SDTP without any weight sharing (i.e. weight transport in the backward pass or weight tying in convolutions) perform much worse than DTP, likely because of impoverished output targets. We also studied an approach to rescue performance with SDTP. Overall, while some variants of TP and FA came close to matching the performance of BP on MNIST and CIFAR, all of the biologically motivated algorithms performed much worse than BP in the context of ImageNet. Our experiments are far from exhaustive and we hope that researchers in the field may coordinate to study the performance of other recently introduced biologically motivated algorithms, including e.g. [28, 27].

We note that although TP and FA algorithms go a long way towards biological plausibility, there are still many biological constraints that we did not address here. For example, we've set aside the question of spiking neurons entirely to focus on asking whether variants of TP can scale up to solve difficult problems *at all*. The question of spiking networks is an important one [35, 12, 7, 34], but it should nevertheless be possible to gain algorithmic insight to the brain without tackling all of the elements of biological complexity simultaneously. Similarly, we also ignore Dale's law in all of our experiments [29]. In general, we've aimed at the simplest models that allow us to address questions around (1) *weight sharing*, and (2) *the form and function of feedback communication*. However, it is worth noting that our work here ignores one other significant issue with respect to the plausibility of feedback communication: BP, FA, all of the TP variants, and indeed most known activation propagation algorithms (for an exception see Sacramento et al. [34]), still require distinct forward and backward (or "positive" and "negative") *phases*. The way in which forward and backward pathways in the brain interact is not well characterized, but we're not aware of existing evidence that straightforwardly supports distinct phases.

Nevertheless, algorithms that aim to illuminate learning in cortex should be able to perform well on difficult domains without relying on any form of weight sharing. Thus, our results offer a new benchmark for future work looking to evaluate the effectiveness of biologically plausible algorithms in more powerful architectures and on more difficult datasets.

#### Acknowledgments

We would like to thank Shakir Mohamed, Wojtek Czarnecki, Yoshua Bengio, Rafal Bogacz, Walter Senn, Joao Sacramento, James Whittington, and Benjamin Scellier for useful discussions.

## Footnotes

[1]`https://github.com/donghyunlee/dtp/blob/master/conti_dtp.py`

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
