[Supplementary Material]

# 5 Appendix

## 5.1 SDTP and AO-SDTP algorithm details

In this section, we provide detailed algorithm description for both SDTP and its extension AO-SDTP which can be found in Algorithm Box 1 and 2.

In the original DTP algorithm, autoencoder training is done via a noise-preserving loss. This is a well principled choice for the algorithm on a computer [21], and our experiments with DTP use this noise-preserving loss. However, in the brain, autoencoder training would necessarily be de-noising, since uncontrolled noise is added downstream of a given layer (e.g. by subsequent spiking activity and stochastic vesicle release). Therefore, in our experiments with SDTP and AO-SDTP we use de-noising autoencoder training.

---

**Algorithm 2** Augmented Output Simplified Difference Target Propagation

---

Propagate activity forward:
**for** $l = 1$ **to** $L$ **do**
  $h_l \leftarrow f_l(h_{l-1}; \theta_l)$
**end for**
Split network output: $[o, z] \leftarrow h_L$
Compute first target: $\hat{o} \leftarrow \mathrm{argmin}_o \mathcal{L}(o)$
Compute target for the penultimate layer: $\hat{h}_{L-1} \leftarrow h_{L-1} - g_L(o, z; \lambda_L) + g_L(\hat{o}, z; \lambda_L)$
Compute targets for lower layers:
**for** $l = L - 2$ **to** $1$ **do**
  $\hat{h}_l \leftarrow h_l - g(h_{l+1}; \lambda_{l+1}) + g(\hat{h}_{l+1}; \lambda_{l+1})$
**end for**
Train inverse function parameters:
**for** $l = L$ **to** $2$ **do**
  Generate corrupted activity $\tilde{h}_{l-1} = h_{l-1} + \epsilon, \epsilon \sim \mathcal{N}(0, \sigma^2)$
  Update parameters $\lambda_l$ using SGD on loss $\mathcal{L}_l^{inv}(\lambda_l)$
  $\mathcal{L}_l^{inv}(\lambda_l) = \|h_{l-1} - g(f(\tilde{h}_{l-1}; \theta_{l-1}); \lambda_l)\|_2^2$
**end for**
Train feedforward function parameters:
**for** $l = 1$ **to** $L$ **do**
  Update parameters $\theta_l$ using SGD on loss $\mathcal{L}_l(\theta_l)$
  $\mathcal{L}_l(\theta_l) = \|f(h_l; \theta_l) - \hat{h}_{l+1}\|_2^2$ if $l < L$, else $\mathcal{L}_L(\theta_L) = \mathcal{L}$ (task loss)
**end for**

---

One might expect the performance of AO-SDTP to depend on the size and the structure of the auxiliary output. We investigated the effect of the auxiliary output size. The results are consistent with the intuition that larger sizes generally lead to better performance, with improvements leveling off once the output is large enough to encode information about the penultimate layer well (see Figure 4).

## 5.2 Architecture details for all experiments

In this section we provide details on the architectures used across all experiments. The detailed specifications can be found in Table 3.

All locally-connected architectures consist of a stack of locally-connected layers (each specified by: receptive field size, number of output channels, stride), followed by one or more fully-connected layers and an output softmax layer. All locally-connected layers use zero padding to ensure unchanged shape of the output with stride $= 1$. One of our general empirical findings was that pooling operations are not very compatible with TP and are better to be replaced with strided locally-connected layers.

The locally-connected architecture used for the ImageNet experiment was inspired by the ImageNet architecture used in [37]. Unfortunately, the naive replacement of convolutional layers with locally-connected layers would result in a computationally-prohibitive architecture, so we decreased number of output channels in the layers and also removed layers with $1 \times 1$ filters. We also slightly decreased filters in the first layer, from $11 \times 11$ to $9 \times 9$. Finally, as in the CIFAR experiments, we replaced

Table 3: Architecture specification. The format for locally-connected layers is (kernel size, number of output channels, stride).

| Dataset | Fully-connected network | Locally-connected network |
|---------|-------------------------|---------------------------|
| MNIST | FC 256<br>FC 256<br>FC 256<br>FC 256<br>FC 256<br>Softmax 10 | $(3 \times 3, 32, 2)$<br>$(3 \times 3, 64, 2)$<br>FC 1024<br>Softmax 10 |
| CIFAR | FC 1024<br>FC 1024<br>FC 1024<br>Softmax 10 | $(5 \times 5, 64, 2)$<br>$(5 \times 5, 128, 2)$<br>$(3 \times 3, 256)$<br>FC 1024<br>Softmax 10 |
| IMAGENET | – | $(9 \times 9, 48, 4)$<br>$(3 \times 3, 48, 2)$<br>$(5 \times 5, 96, 1)$<br>$(3 \times 3, 96, 2)$<br>$(3 \times 3, 192, 1)$<br>$(3 \times 3, 192, 2)$<br>$(3 \times 3, 384, 1)$<br>Softmax 1000 |

all pooling operations with strided locally-connected layers and completely removed the spatial averaging in the last layer that we previously found problematic when learning with TP.

## 5.3 Details of hyperparameter optimization

For DTP and SDTP we optimized over parameters of: (1) the forward model and inverse Adam optimizers, (2) the learning rate $\alpha$ used to compute targets for $h_{L-1}$ in DTP, and (3) the Gaussian noise magnitude $\sigma$ used to train inverses. For backprop we optimized only the forward model Adam optimizer parameters. For all experiments the best hyperparameters were found by random searches over 60 random configurations drawn from the ranges specified in table 4. We provide values for the best configurations in table 5.

As we pointed out in section 3, the explored learning methods have different sensitivity to hyperparameters. We provide histograms of the best test accuracies reached by different hyperparameter configurations on MNIST and CIFAR for each of the experiments (see Figure 5). We were not able

Figure 4: Auxiliary output size effect on CIFAR performance.

(a) MNIST, fully-connected networks.

(b) MNIST, locally-connected networks.

(c) CIFAR, fully-connected networks.

(d) CIFAR, locally-connected networks.

Figure 5: Distribution of test accuracies achieved under different hyperparameters.

Table 4: Hyperparameter search space used for the experiments

| HYPERPARAMETER | SEARCH DOMAIN |
|---|---|
| Learning rate of model Adam optimizer | $[10^{-5}; 3 \times 10^{-4}]$ |
| $\beta_1$ parameter of model Adam optimizer | Fixed to 0.9 |
| $\beta_2$ parameter of model Adam optimizer | $\{0.99, 0.999\}$ |
| $\epsilon$ parameter of model Adam optimizer | $\{10^{-4}, 10^{-6}, 10^{-8}\}$ |
| Learning rate of inverse Adam optimizer | $[10^{-5}; 3 \times 10^{-4}]$ |
| $\beta_1$ parameter of inverse Adam optimizer | Fixed to 0.9 |
| $\beta_2$ parameter of inverse Adam optimizer | $\{0.99, 0.999\}$ |
| $\epsilon$ parameter of inverse Adam optimizer | $\{10^{-4}, 10^{-6}, 10^{-8}\}$ |
| Learning rate $\alpha$ used to compute targets for $h_{L-1}$ in DTP | $[0.01; 0.2]$ |
| Gaussian noise magnitude $\sigma$ used to train inverses | $[0.01; 0.3]$ |

to collect the results for each of the exploratory runs on the ImageNet due to prohibitive demand on computation. In this case, we also started 60 random configurations but after 10 epochs we allowed only the best performing job to continue thereafter.

These experiments demonstrate clearly that BP is the most stable algorithm. TP methods proved to be the most sensitive to the choice of hyperparameters, likely due to complicated interactions between updating forward and inverse weights. Finally, we note that within the TP based method, the alternating update schedule not only reach the better accuracy, but overall led to more stable convergence.

## 5.4 Implementation details for locally-connected architectures

Although locally-connected layers can be seen as a simple generalization of convolution layers, their implementation is not entirely straightforward. First, a locally-connected layer has many more trainable parameters than a convolutional layer with an equivalent specification (i.e. receptive field size, stride and number of output channels). This means that a simple replacement of every convolutional layer with a locally-connected layer can be computationally prohibitive for larger networks. Thus, for large networks, one has to decrease the number of parameters to run experiments using a reasonable amount of memory and compute. In our experiments we opted to decrease the number of output channels in each layer by a given factor. Obviously, this can have a negative effect on the resulting performance and more work needs to be done to scale locally-connected architectures.

**Inverse operations**   When training locally-connected layers with target propagation, one also needs to implement the inverse computation in order to train the feedback weights. As in fully-connected layers, the forward computation implemented by both locally-connected and convolutional layers can be seen as a linear transformation $y = Wx + b$, where the matrix $W$ has a special, sparse structure (i.e., has a block of non-zero elements, and zero-elements elsewhere), and the dimensionality of $y$ is not more than $x$.

The inverse operation requires computation of the form $x = Vy + c$, where matrix $V$ has the same sparse structure as $W^T$. However, given the sparsity of $V$, computing the inverse of $y$ using $V$ would be highly inefficient [9]. We instead use an implementation trick often applied in deconvolutional architectures. First, we instantiate a forward locally-connected computation $z = Ax$, where $z$ and $A$ are dummy activities and sparse weights. We then express the transposed weight matrix as the *gradient* of this feedforward operation:

$$V = A^T = \left(\frac{dz}{dx}\right)^T, \text{ and thus } x = Vy + c = \left(\frac{dz}{dx}\right)^T y + c.$$

The gradient $\frac{dz}{dx}$ (and its multiplication with $y$) can be very quickly computed by the means of automatic differentiation in many popular deep learning frameworks. Hence one only needs to define the forward locally-connected computation and the corresponding transposed operation is implemented trivially. Note that this is strictly an implementation detail and does not introduce any additional use of gradients or weight sharing in learning.

Table 5: Best hyperparameters found by the random search.

### MNIST, Fully-connected

| | | DTP PARALLEL | DTP ALTERNATING | SDTP PARALLEL | SDTP ALTERNATING | BP | BP ConvNet | FA | DFA |
|---|---|---|---|---|---|---|---|---|---|
| MODEL | LR | 0.000757 | 0.000308 | 0.000402 | 0.000301 | 0.000152 | | 0.000168 | 0.001649 |
| | $\beta_1$ | 0.99 | 0.99 | 0.99 | 0.9 | 0.9 | | 0.9 | 0.9 |
| | $\beta_2$ | 0.95 | 0.99 | 0.999 | 0.95 | 0.999 | | 0.999 | 0.95 |
| | $\epsilon$ | $10^{-3}$ | $10^{-4}$ | $10^{-8}$ | $10^{-4}$ | $10^{-8}$ | | $10^{-4}$ | $10^{-3}$ |
| INVERSE | LR | 0.000768 | 0.004593 | 0.001101 | 0.009572 | | | | |
| | $\beta_1$ | 0.99 | 0.99 | 0.99 | 0.9 | | | | |
| | $\beta_2$ | 0.999 | 0.999 | 0.95 | 0.95 | | | | |
| | $\epsilon$ | $10^{-4}$ | $10^{-4}$ | $10^{-6}$ | $10^{-3}$ | | | | |
| | $\alpha$ | 0.15008 | 0.231758 | | | | | | |
| | $\sigma$ | 0.36133 | 0.220444 | 0.213995 | 0.118267 | | | | |

### MNIST, Locally-connected

| | | DTP PARALLEL | DTP ALTERNATING | SDTP PARALLEL | SDTP ALTERNATING | BP | BP ConvNet | FA | DFA |
|---|---|---|---|---|---|---|---|---|---|
| MODEL | LR | 0.000905 | 0.001481 | 0.000145 | 0.000651 | 0.000133 | 0.000297 | 0.000219 | 0.002462 |
| | $\beta_1$ | 0.9 | 0.9 | 0.9 | 0.9 | 0.9 | 0.9 | 0.9 | 0.9 |
| | $\beta_2$ | 0.99 | 0.99 | 0.99 | 0.99 | 0.99 | 0.99 | 0.999 | 0.99 |
| | $\epsilon$ | $10^{-4}$ | $10^{-4}$ | $10^{-6}$ | $10^{-4}$ | $10^{-8}$ | $10^{-8}$ | $10^{-6}$ | $10^{-4}$ |
| INVERSE | LR | 0.001239 | 0.000137 | 0.001652 | 0.003741 | | | | |
| | $\beta_1$ | 0.9 | 0.9 | 0.9 | 0.9 | | | | |
| | $\beta_2$ | 0.999 | 0.999 | 0.999 | 0.99 | | | | |
| | $\epsilon$ | $10^{-4}$ | $10^{-6}$ | $10^{-4}$ | $10^{-4}$ | | | | |
| | $\alpha$ | 0.116131 | 0.310892 | | | | | | |
| | $\sigma$ | 0.099236 | 0.366964 | 0.061555 | 0.134739 | | | | |

### CIFAR, Fully-connected

| | | DTP PARALLEL | DTP ALTERNATING | SDTP PARALLEL | SDTP ALTERNATING | BP | BP ConvNet | FA | DFA |
|---|---|---|---|---|---|---|---|---|---|
| MODEL | LR | 0.000012 | 0.000013 | 0.000129 | 0.000041 | 0.000019 | | 0.000025 | 0.000050 |
| | $\beta_1$ | 0.9 | 0.9 | 0.9 | 0.9 | 0.9 | | 0.9 | 0.9 |
| | $\beta_2$ | 0.999 | 0.999 | 0.99 | 0.99 | 0.999 | | 0.99 | 0.99 |
| | $\epsilon$ | $10^{-8}$ | $10^{-8}$ | $10^{-6}$ | $10^{-4}$ | $10^{-6}$ | | $10^{-4}$ | $10^{-8}$ |
| INVERSE | LR | 0.000039 | 0.000114 | 0.000011 | 0.000014 | | | | |
| | $\beta_1$ | 0.9 | 0.9 | 0.9 | 0.9 | | | | |
| | $\beta_2$ | 0.99 | 0.99 | 0.99 | 0.99 | | | | |
| | $\epsilon$ | $10^{-6}$ | $10^{-4}$ | $10^{-6}$ | $10^{-8}$ | | | | |
| | $\alpha$ | 0.125693 | 0.172085 | | | | | | |
| | $\sigma$ | 0.169783 | 0.134811 | 0.273341 | 0.125678 | | | | |

### CIFAR, Locally-connected

| | | DTP PARALLEL | DTP ALTERNATING | SDTP PARALLEL | SDTP ALTERNATING | BP | BP ConvNet | FA | DFA |
|---|---|---|---|---|---|---|---|---|---|
| MODEL | LR | 0.000032 | 0.000036 | 0.000020 | 0.000109 | 0.000044 | 0.000133 | 0.000022 | 0.000040 |
| | $\beta_1$ | 0.9 | 0.9 | 0.9 | 0.9 | 0.9 | 0.9 | 0.9 | 0.9 |
| | $\beta_2$ | 0.99 | 0.99 | 0.99 | 0.99 | 0.999 | 0.99 | 0.999 | 0.999 |
| | $\epsilon$ | $10^{-6}$ | $10^{-8}$ | $10^{-4}$ | $10^{-4}$ | $10^{-6}$ | $10^{-4}$ | $10^{-8}$ | $10^{-8}$ |
| INVERSE | LR | 0.000852 | 0.000389 | 0.000261 | 1.1e-05 | | | | |
| | $\beta_1$ | 0.9 | 0.9 | 0.9 | 0.9 | | | | |
| | $\beta_2$ | 0.999 | 0.999 | 0.999 | 0.99 | | | | |
| | $\epsilon$ | $10^{-4}$ | $10^{-8}$ | $10^{-6}$ | $10^{-8}$ | | | | |
| | $\alpha$ | 0.189828 | 0.208141 | | | | | | |
| | $\sigma$ | 0.146728 | 0.094869 | 0.299769 | 0.023804 | | | | |

### ImageNet, Locally-connected

| | | DTP PARALLEL | DTP ALTERNATING | SDTP PARALLEL | BP | BP ConvNet | FA |
|---|---|---|---|---|---|---|---|
| MODEL | LR | 0.000217 | 0.000101 | 0.000011 | 0.000024 | 0.000049 | 0.000043 |
| | $\beta_1$ | 0.9 | 0.9 | 0.9 | 0.9 | 0.9 | 0.9 |
| | $\beta_2$ | 0.99 | 0.99 | 0.999 | 0.999 | 0.99 | 0.99 |
| | $\epsilon$ | $10^{-6}$ | $10^{-8}$ | $10^{-6}$ | $10^{-8}$ | $10^{-8}$ | $10^{-4}$ |
| INVERSE | LR | 0.000234 | 0.000064 | 0.000170 | | | |
| | $\beta_1$ | 0.9 | 0.9 | 0.9 | | | |
| | $\beta_2$ | 0.999 | 0.999 | 0.999 | | | |
| | $\epsilon$ | $10^{-6}$ | $10^{-8}$ | $10^{-8}$ | | | |
| | $\alpha$ | 0.163359 | 0.03706 | | | | |
| | $\sigma$ | 0.192835 | 0.097217 | 0.168522 | | | |

Figure 6: Train (solid) and test (dashed) reconstruction errors on MNIST.

Figure 7: MNIST reconstructions obtained by different learning methods. Even though SDTP produces more artifacts, the visual quality is comparable due to the presence of diverse targets.

## 5.5 Autoencoding and target diversity

Since one of the main limitations of SDTP is target diversity for the penultimate layer, it may be instructive to compare different learning methods on a task that involves rich output targets. A natural choice for such a task is learning an autoencoder with the reconstruction error as a loss.

We set a simple fully-connected architecture for the autoencoder of the following structure $28 \times 28 - 512 - 64 - 512 - 28 \times 28$ and trained it on MNIST using squared $L_2$ reconstruction error. The training curves can be found in Figure 6. SDTP still demonstrated a tendency to underfit, and did not match performance of DTP and backpropagation. But, visual inspection of reconstructions on the test set of MNIST did not show a significant difference in the quality of reconstructions (see Figure 7), which supports the hypothesized importance of target diversity for SDTP performance.

Figure 8: Train (dashed) and test (solid) classification errors on MNIST.

## 5.6  Backpropagation as a special case of target propagation

Even though difference target propagation is often contrasted with back-propagation, it is interesting to note that these procedures have a similar functional form. One can ask the following question: that should the targets be in order to make minimization of the learning loss equivalent to a backpropagation update?

Formally, we want to solve the following equation for $\hat{h}_l$:

$$\frac{d}{d\theta_l}\frac{\mathcal{L}_l}{2} = \frac{d}{d\theta_l}\mathcal{L}.$$

Here we divided the learning loss by 2 to simplify the following calculations. Transforming both sides of the equation, we obtain

$$\frac{dh_l}{d\theta_l}(h_l - \hat{h}_l) = \frac{dh_l}{d\theta_l}\frac{d\mathcal{L}_y}{dh_l},$$

from which it follows that

$$\frac{d\mathcal{L}_y}{dh_l} = h_l - \hat{h}_l \text{ and } \hat{h}_l = h_l - \frac{d\mathcal{L}_y}{dh_l}.$$

We now expand the latter equation to express $\hat{h}_l$ through $\hat{h}_{l+1}$:

$$\hat{h}_l = h_l - \frac{dh_{l+1}}{dh_l}\frac{d\mathcal{L}_y}{dh_{l+1}} = h_l - \frac{dh_{l+1}}{dh_l}(h_{l+1} - \hat{h}_{l+1}).$$

Finally, if we define $g(\tilde{h}_{l+1}) = g_{bp}(\tilde{h}_{l+1}) = \frac{dh_{l+1}}{dh_l}\tilde{h}_{l+1}$, then a step on the local learning loss in TP will be equivalent to a gradient descent step on the global loss.

The question remains whether this connection might be useful for helping us to think about new learning algorithms. For example, one could imagine an algorithm that uses hybrid targets, e.g. computed using a convex combination of the differential and the pseudo-inverse $g$-functions:

$$g(\tilde{h}_{l+1}) = \alpha g_{bp}(\tilde{h}_{l+1}) + (1-\alpha)g_{tp}(\tilde{h}_{l+1}), \quad 0 \le \alpha \le 1.$$

Continuing the analogy between these two methods, is it possible that the inverse loss could be a useful regularizer when used with $g_{bp}$? Practically that would mean that we want to regularize parameters of the forward computation $f$ indirectly through its derivatives. Interestingly, in the one-dimensional case (where $h_l$ and $f(h_l)$ are scalars) the inverse loss is minimized by $f(h_l) = \pm\sqrt{h_l^2 + b}$.