[Reviews · NeurIPS 2018]

Reviewer 1



The authors provide a clear and succinct introduction to the problems and approaches of biologically plausible forms of backprop in the brain. They argue for behavioural realism apart from physiological realism and undertake a detailed comparison of backprop versus difference target prop and its variants (some of which they newly propose) and also direct feedback alignment. In the end though, they find that all proposed forms of bio-plausible alternatives to backprop fall quite short on complex image recognition tasks. Despite the negative results, I find such a comparison very timely to consolidate results and push the community to search for better and more diverse alternatives. Overall I find the work impressive. Just a few comments. 1. The authors claim that weight sharing is not plausible in the brain. If one approaches from the point of view of propagating errors back, yes weight sharing is difficult to imagine. But from the point of view of unsupervised learning, if each local patch receives similar local statistics of images, it is conceivable that all the different local patches of neurons will end up responding to similar features, effectively having similar weights. Thus one might wish to train initial layers with some form of weight sharing and only train the output layers with forms of backprop, TP etc. Further in the discussion the authors claim: " we’ve aimed at the simplest models that allow us to address questions around (1) weight sharing, ". However, it seems to me that the authors actually did not implement weight sharing at all in the networks. This should be clarified. 2. The authors only consider image recognition tasks and it is conceivable that here the network structure matters far more than the learning rule. For example the brain may use some form of recurrence for pattern completion and might also build internal 3D models of objects from the images. For later, possibly other tasks should be also tested, but here at least this lacuna should be emphasized in the discussion. Minor: l 68: "is unlikely unveil" Overall, if the authors addressed these points, I think that the paper is suited for publication at NIPS. --------------------------------------------- After reading the other reviews and the author rebuttal, and after discussions with other reviewers on the forum, I still feel that the work is impressive, but perhaps not in the top 50% of accepted NIPS papers. Also, the results may not be conclusive since the training was only for a fixed number of epochs and test error was still decreasing in Fig 1 (right) for some variants. ---------------------------------------------

Reviewer 2



Summary Under the objective of finding biologically plausible alternatives to backpropagation that perform well on difficult tasks, variants of difference target propagation are compared to direct feedback alignment and backpropagation. All variants are found to perform significantly worse than backpropagation in digit classification (MNIST) and, in particular, in object classification (CIFAR-10 and ImageNet). Quality The study is mostly well done and the results seem plausible. But, to qualify as a solid and reproducible benchmark, I expect more details. Also, some parts feel a bit preliminary. 1. For how many epochs were the different methods trained? Do you report optimal values for a fixed number of training epochs or do you report the test error at the optimal validation value for each method? It seems, for example, that the test classification errors in Figure 2 (right) are still decreasing after epoch 500 for all variants of DTP. 2. With how many different initialisations where the results obtained in table 1 and 2? Did you repeat each experiment so many times that confidence interval are negligible? 3. Can you report the best hyperparameters in addition to the details of hyperparameter optimization? 4. AO-SDTP was tested only for 512 auxiliary outputs and the authors refer to future work (line 268). It would be interesting to know more about the performance of AO-SDTP for different dimensionality of z; potentially even for different ways to obtain the random features. 5. Since DTP alternating was always best on MNIST and CIFAR-10, I think it would be interesting to see it's performance on ImageNet, also to have an estimate of what ideally could be achieved with AO-SDTP. Clarity The paper is well written and easy to follow. One minor point minor point where I stumbled, though, was in table 2, where top-1 and top-5 is not defined. For readers familiar with ILSVRC2012 - I guess this is what you are using - it is probably immediately clear what is meant. For everybody else, one sentence of explanation would help. Another minor point that I believe could still be improved is the notation of the different losses. For example, why L(h_L) and not L(Theta)? What is L_y in section 5.5 of the appendix? Originality The work is original in the sense that the evaluation of DTP variants on MNIST, CIFAR-10 and Imagenet was never published before. The variants themselves are original as well, but minor and straightforward modifications of the original DTP. Significance It is important that people working on biologically plausible alternatives to backpropagation know about the limitations of DTP and variants. But since the paper does not present an advancement of state-of-the-art alternatives to backpropagation I think it would be better suited for a more specialized audience, maybe at a NIPS workshop. Minor Points - In Figure 2 & 4 the colours for parallel and alternating are hard to distinguish in print. - References 12 & 13 (identical up to title) could be replaced by https://elifesciences.org/articles/22901 - line 47: connections - line 68: to unveil - lines 77-78: I am not a native English speaker but "which eliminate significant lingering biologically implausible features" sounds strange to me. - lines 109-110: I don't think that the backprop equations themselves imply a mode of information propagation that does not influence neural activity. The neural activity in different phases could corresponds to different quantities of the backprop algorithm. - I don't understand the sentence that starts on line 182. - line 259: perform better - I think the statement starting on line 294 is a bit strong. Reference 37 falls in this period and I am sure one can find further works in computational neuroscience on this topic between the mid 1990s and 2010. Let me conclude with two questions that go beyond the discussion of this specific paper but address the attempt to find biologically plausible alternatives to backpropagation more broadly. I agree with the authors that biologically plausible methods should also be evaluated on behavioural realism. But what is a behaviourally realistic task? Is the repetitive presentation of static images in minibatches together with corresponding labels a behaviourally realistic task? Shouldn't we rather search for difficult and behaviourally realistic tasks that are less inspired by machine learning challenges but more by actual tasks humans solve? ==== The author's response answered most questions and concerns. Suboptimal I find the decision to compare the methods on a fixed number of epochs. I can understand it because of limited computation resources. But since some of the curves did not seem to saturate (see point 1 above) I would be curious to know whether the bio-plausible alternatives would become significantly better if the number of epochs were a tunable hyper-parameter. Note that I think optimal performance after a fixed number of epochs is a perfect metric in machine learning but I am not convinced it is the right one to compare bio-plausible alternatives, since our random initializations of the network weights probably don't capture the strong inductive biases that evolutionary processes instantiate in biological neural networks.

Reviewer 3



This well-written paper explores and extends “Difference Target Propagation” (DTP), a non-backprop deep learning algorithm for artificial neural networks. The authors suggest two modifications to DTP to make it more biologically plausible, and another modification to improve performance when the targets are class labels. Experiments compare DTP and the modified versions to BP and another algorithm, DFA, on MNIST, CIFAR-10, and ImageNet. All three of the proposed modifications to the DTP algorithm seem incremental, but they are very reasonable and the experimental results support the hypotheses. The analysis and results are nicely presented. The other contribution is the experimental results on large data sets. The authors falsely claim that this is the first study to apply biologically-motivated learning methods to CIFAR-10 (Lines 243-245), they should cite Nokland “Direct feedback alignment provides learning in deep neural networks” and Baldi et al. “Learning in the machine: Random backpropagation and the deep learning channel,” both of which experiment with FA and DFA on CIFAR-10. As far as I know, this is the first attempt using such algorithms on ImageNet, but as the authors report, it is difficult get good performance on large problems using new learning algorithms, as hyperparameters need to be carefully re-tuned and the network might need to be larger.